# The Affective Dimension of Pain Appears to Be Determinant within a Pain–Insomnia–Anxiety Pathological Loop in Fibromyalgia: A Case-Control Study

**DOI:** 10.3390/jcm11123296

**Published:** 2022-06-08

**Authors:** Lliure-Naima Mory, Daniel de Oliveira Fernandes, Christian Mancini, Michael Mouthon, Joelle Nsimire Chabwine

**Affiliations:** 1Neurology Unit, Medicine Section, Department of Neuroscience and Movement Science, Faculty of Science and Medicine, University of Fribourg, 1700 Fribourg, Switzerland; naima.mory@gmail.com (L.-N.M.); daniel.deoliveirafernandes@unifr.ch (D.d.O.F.); christian.mancini@hotmail.com (C.M.); michael.mouthon@unifr.ch (M.M.); 2Division of Neurorehabilitation, Fribourg Cantonal Hospital, 1700 Fribourg, Switzerland

**Keywords:** fibromyalgia, affective pain, insomnia, anxiety, ongoing pain

## Abstract

Background: Fibromyalgia (FM) is a chronic pain disease characterized by multiple symptoms whose interactions and implications in the disease pathology are still unclear. This study aimed at investigating how pain, sleep, and mood disorders influence each other in FM, while discriminating between the sensory and affective pain dimensions. Methods: Sixteen female FM patients were evaluated regarding their pain, while they underwent—along with 11 healthy sex- and age-adjusted controls—assessment of mood and sleep disorders. Analysis of variance and correlations were performed in order to assess group differences and investigate the interactions between pain, mood, and sleep descriptors. Results: FM patients reported the typical widespread pain, with similar sensory and affective inputs. Contrary to controls, they displayed moderate anxiety, depression, and insomnia. Affective pain (but neither the sensory pain nor pain intensity) was the only pain indicator that tendentially correlated with anxiety and insomnia, which were mutually associated. An affective pain–insomnia–anxiety loop was thus completed. High ongoing pain strengthened this vicious circle, to which it included depression and sensory pain. Conclusions: Discriminating between the sensory and affective pain components in FM patients disclosed a pathological loop, with a key role of affective pain; high ongoing pain acted as an amplifier of symptoms interaction. This unraveled the interplay between three of most cardinal FM symptoms; these results contribute to better understand FM determinants and pathology and could help in orienting therapeutic strategies.

## 1. Introduction

Fibromyalgia (FM) is characterized by a constellation of symptoms including mainly chronic widespread pain, sleep disorders, fatigue, mood dysfunction, and cognitive impairment [1,2]. This highly disabling disease is prevalent in the general population (0.2–6.6%) and predominates among women [3]. It is unclear whether nonpain FM cardinal symptoms (e.g., mood and sleep disorders) intrinsically participate in the disease’s pathological processes or simply act as comorbidities. Furthermore, how their mutual negative influence on one another could contribute to the disease pathology has yet to be enlightened.

Sleep disorders are present in ~90% of FM patients [4,5,6] and are closely associated with chronic pain. People suffering from insomnia witness, similar to healthy individuals under sleep deprivation [7,8], lower pain thresholds [9] and myalgia (recalling FM symptoms), while chronic pain gives rise to sleep disturbances [10]. Hence, this bidirectional relation between chronic pain (including in FM) and sleep disorders creates a vicious circle [4,11]. However, it is not established whether there could be a predominant direction in this relationship [12,13]. On the other hand, in FM patients, sleep disorders seem to be related to mood dysfunction (anxiety in particular), in addition to other symptoms such as memory and concentration difficulties [6]. Furthermore, prevalent anxiety (60%) and depression (20–80%) in FM [14] are associated with pain severity [15,16,17]. Thus, from a purely clinical perspective, chronic pain, mood, and sleep disorders seem to be all associated one with another, while the primum movens, as well as the preferential direction within these intricated associations, remain unclear owing to contradictory observations [18,19].

Although the etiology and pathological mechanisms of FM are unknown, the abovementioned multiple interactions between pain, sleep, and mood disorders could relate to FM-induced central nervous system aberrant functioning. Indeed, brain areas and neurotransmitter pathways involved in pain, sleep, and mood disorders overlap [20,21,22]. Thus, a better understanding of the complex interplay between these symptoms would not only give new insights into elucidating pathological processes engaged in FM but would also further enlighten underlying neurophysiological and structural mechanisms.

Difficulties understanding the interplay between pain, mood, and sleep disorders in FM might be due to the multiple dimensions of pain, making it difficult to linearly establish a relationship with other symptoms. In fact, most studies solely evaluate pain intensity [23], while a better approach would consist, for example, in differentiating between the sensory and the affective components of pain [24,25]. The sensory-discriminative dimension refers to the spatiotemporal aspects and the quality of pain, while the affective dimension states the unpleasantness and motivating behaviors such as avoidance or escape. These two pain modalities are believed to be partially independent one from another [26,27], while their relationship with pain intensity is not clearly studied. Neuroanatomically, brain areas involved in cognitive–emotional regulation overlap with those processing the affective dimension of pain (e.g., the limbic system), whereas structures such as the primary and secondary somatosensory cortices and the thalamus are more specifically involved in sensory dimension pain [28].

There are indications that sensory and affective pain components could be differently associated with the disease outcome or with other symptoms. For instance, in orofacial chronic pain, the sensory pain level predicts low functional outcome but high social support, while affective pain predicts higher pain interference and affective distress [29]. Moreover, affective pain can be selectively altered by mood [30,31] and by cognitive manipulations [32]. Further, there exists evidence that the association between pain and sleep disorders probably relies on neurophysiological alterations such as increased pain facilitation or impaired pain inhibition [11]. Finally, FM and mood disorders share similar stress-induced pathophysiologies [33].

Affective and sensory pain dimensions are partially independent [27,34], implying that mechanisms linking them to other clinical symptoms such as sleep and mood disorders would not be identical. Knowing the prominent affective dysfunction observed in FM [33], we made the general hypothesis that the affective component of pain would play a predominant role in the interplay between pain, mood disorders, and sleep dysfunction, thereby showing a stronger association with them than the sensory component of pain and other pain indicators. The aim of this study was, therefore, to investigate how pain (evaluated through its affective and sensory dimensions, as well as other indicators), mood, and sleep disorders interact in FM patients.

## 2. Materials and Methods

### 2.1. Study Design and Ethics

This case-control observational study involved patients suffering from FM and healthy controls of similar age. Ethical clearance was obtained from the Ethical Committee of Vaud (CER-VD) under the number PB_2016-00739 (initial number 331/15). Each participant signed an informed consent form prior to any data collection and received a financial compensation thereafter. The study complied with all international ethical regulations, including the Helsinki declaration.

### 2.2. Participants

Patients were recruited between November 2018 and February 2020, mainly through neurologists, rheumatologists, and pain specialists from Fribourg Hospital, and through Swiss FM associations, using web-based, oral, and flyer advertisements. The diagnosis of FM had to be made by a specialist (rheumatologist, neurologist, or pain specialist) and meet internationally admitted diagnosis criteria (see below). All participants were adults (age ≥ 18 years old) and right-handed, following indications for possible lateralization in brain function related to chronic pain [35], including in FM [36]. In order to avoid data variability due to sex influence on pain perception [37,38] and taking into account the epidemiology of FM [39], we included exclusively females. Exclusion criteria for all participants consisted of the following: existence of central nervous system lesion or diseases such as epilepsy or parasomnia, severe cognitive impairment or psychiatric disease, and surgery involving any nervous system structure less than six months before inclusion. Existence of any pain was an additional exclusion criterion in controls. In total, 16 FM patients and 11 controls (aged 51.8 (8.5) and 54.2 (4.6) years, respectively) were included in the final analysis of the study, as shown in the selection flowchart (Figure 1).

### 2.3. Data Collection

Data were collected in the frame of a larger study investigating the involvement of brain GABAergic signaling in FM. However, in this paper, we exclusively focused on results related to clinical pain, sleep, and mood disorders assessments. Each participant was interviewed following a standardized questionnaire (general epidemiological data, treatments, relevant medical history) and underwent a brief neurological examination for further characterization in order to identify exclusion criteria.

The Insomnia Severity Index (ISI) was compiled for insomnia assessment [40] and the Hospital Anxiety and Depression Scale (HADS) was employed to determine the existence of anxiety and depression [41]. Three specific questionnaires (the Fibromyalgia Rapid Screening Tool (FiRST) [42], the Symptoms Severity Score (SSS), and the Widespread Pain Index (WPI) of the 2010 American College of Rheumatology criteria (ACR 2010)) were used for the diagnosis of FM. The Visual Analogue Scale (VAS) was used to quantify pain intensity on the day of evaluation and the average pain level over the week prior [43]; the Douleur Neuropathique 4 questionnaire (DN4) to assess the neuropathic pattern of pain [44]; the Short-form McGill Pain Questionnaire to separately evaluate sensory (MPQsens) and affective (MPQaff) pain components [45] and the obtained scores were reported out of 10 for easy interpretation in comparison with the VAS. Within the Brief Pain Inventory (BPI), the impact of pain on patients’ life was assessed using the BPI severity (BPIsev) and BPI interference (BPIint) composite scores [46]. The WPI was further considered for a quantitative estimate of pain extension over the body in order to assess how this factor could relate to other clinical indicators.

To determine pathological mood and sleep scores, the following cut-offs were considered: HADanx/HADdep, respectively: <7 = normal mood state, 8–10 = mild anxiety/depression, 11–14 = moderate anxiety/depression, 15–21 = severe anxiety/depression [41]; ISI: <7 = no clinically significant insomnia, 8–14 = subthreshold insomnia, 15–21 = clinical insomnia of moderate intensity, 22–28 = clinical insomnia of severe intensity [40]. The neuropathic pattern of pain was confirmed when the DN4 was ≥4 [44].

### 2.4. Data Analysis

#### 2.4.1. FM Diagnosis and Classifications of Clinical Scores

The diagnosis of FM was confirmed by internationally accepted standards using the FiRST score (>5) and the ACR 2010 criteria (i.e., SSS ≥ 7 and WPI ≥ 5 or SSS ≥ 9 and WPI 3–6) [47]. Since pain score cut-offs are highly debated and vary according to studies and contexts, we opted, in general, to analyze pain data without classification or cut-off. We assessed specific descriptors and correlations, except when analyzing the current (ongoing) pain intensity, knowing its emerging role in FM-related pathological modifications occurring in pain matrix [48]. Taking into account that the definition of successful analgesia is VAS < 3 [49] and that our previous data showed significant differences in pain-related neurophysiological EEG markers when VAS ≥ 3 [35], we considered the latter threshold to define a significant pain, even if other classifications of pain severity exist [43,50]. Additionally, pain was considered to be moderate and severe when VAS equaled, respectively, 4.5–7.4 and 7.5–10 [51].

#### 2.4.2. Statistical Analysis

Statistical analyses were performed with R software using standard descriptors (mean (SD)) and significance was admitted at *p* < 0.05. Group differences were evaluated using analysis of variance (ANOVA) and correlations using Pearson’s r correlation coefficient (two-sided). The Bayes factor (BF) was additionally computed for the complementary interpretation of correlation trends in order to estimate potential biases due to the small sample size: while a BF ≥ 3 was in favor of the non-null hypothesis, a BF ≤ 1 was considered as favoring the null hypothesis, and a BF value between 1 and 3 to have undetermined value [52].

## 3. Results

### 3.1. General Data

Patients complained of the typical FM widespread pain (WPI 12.69(3.57)), with moderate pain intensity on the day of evaluation (VASd 4.75(2.84)/10) and higher level during the week before assessment (6.38(2.11), *p* = 0.035). The latter score was similar to the score of the BPI question n°5 recalling the general pain level (6.25(1.61), *p* = 0.797). Thus, we referred to last week’s VAS as VASgen. MPQaff and MPQsens were assessed to be similar (5.88(2.87)/10 and 5.29(2.33)/10, respectively; *p* = 0.318). Patients evaluated their pain as being overall moderately severe (BPIsev 5.97(1.32)/10) and prominently interfering with their daily life (BPIint 6.24(1.50)/10). Most FM patients described their pain in terms of recalling neuropathic features, which was confirmed by the DN4 score of 5.75(1.95). Patients’ sensory impairments and treatments are detailed in Table 1.

Patients suffered from moderate insomnia (68.75% with ISI ≥ 15) [40] and were moderately anxious (62% with HADanx ≥ 8) and depressed (87.5% with HADdep ≥ 8), while controls displayed neither insomnia nor mood disorders (Figure 2).

### 3.2. Associations between Pain Indicators

MPQaff correlated with MPQsens and, in the subgroup of patients undergoing high ongoing pain intensity (VASd ≥ 3), it tended to correlate with VASd (Table 2). VASgen correlated with MPQsens. Beyond the relationship between affective and sensory pain, these observations suggested that affective pain went along with the ongoing pain intensity, while the general pain could be related to the sensory dimension of pain. The DN4 score was more strongly related to MPQsens than MPQaff and correlated to VASgen, while the WPI was solely correlated to VASd. All pain characteristics statistically related to MPQsens were individually correlated to the BPIsev score, although affective pain also displayed a correlation, albeit to a lesser extent, with BPIsev. In contrast, only MPQaff was correlated with BPIint.

Overall, it appeared that sensory pain was linked with specific pain characteristics (neuropathic pain, general pain level), in relation with pain severity, while affective pain, although correlated with sensory pain, interfered more with the patients’ lives.

### 3.3. Associations between Mood, Sleep, and Pain Descriptors

HADanx positively correlated with ISI and with HADdep, while there was no correlation between ISI and HADdep. On the other hand, MPQaff was the only pain score that tended to correlate with both HADanx and ISI, but not with HADdep. No correlation was observed between VASd, VASgen, and MPQsens and any of the mood and sleep indicators (Table 2). When controlling for HADanx, the correlation between ISI and MPQaff decreased and became nonsignificant (r = 0.30, *p* = 0.272), indicating that HADanx influenced the association between ISI and MPQaff. Thus, anxiety could at least partially mediate the link between insomnia and affective pain. When patients were divided into pathological and nonpathological subgroups, respectively, according to the existence of anxiety (HADanx ≥ 8) and insomnia (ISI ≥ 15) (Table 3), anxious and insomniac patients showed significantly higher MPQaff scores than nonaffected patients. Patients displaying clinical insomnia were more significantly anxious than noninsomniac individuals, with no difference noticed in their HADdep scores, while those with clinical anxiety tended to have higher ISI scores than nonanxious patients (significance not reached). Of notice, the abovementioned pathological categorizations did not disclose any difference regarding MPQsens or any of the VAS scores. Comparison between depressed and nondepressed individuals was not possible, as the majority of patients (87.5%) had pathological HADdep scores.

In summary, the affective component of pain appeared to be the only pain descriptor sensitive to the coexistence of anxiety and insomnia in FM patients, in contrast to other pain characteristics, whereas insomnia and anxiety reciprocally influenced each other (with possibly more influence of insomnia on anxiety than the opposite).

### 3.4. Impact of Ongoing Pain

In presence of high ongoing pain (VASd ≥ 3), HADanx was more strongly correlated to ISI but similarly associated to HADdep, which was linked to MPQsens, BPIint, and ISI (Table 2). However, in absence of significant pain intensity, all the abovementioned associations disappeared (data not shown). Overall, the ongoing pain appeared as an “amplifier”, connecting the current pain intensity to the affective pain and strengthening the relationship between sleep and mood dysfunctions.

## 4. Discussion

In this study, we investigated the associations between pain, mood, and sleep disorders in FM patients in order to disentangle the complex interplay between these important clinical factors, with the hypothesis that the affective component of pain would be a key player. To our knowledge, this is the first study investigating the association between pain, mood, and sleep through discrimination between the sensory and affective pain dimensions in FM. However, and despite interesting findings, these results should be considered with caution—due to the small sample size—and further confirmed with larger studies.

While both anxiety and insomnia were tendentially correlated to affective pain, no correlation was observed with other pain indicators. Furthermore, the existence of anxiety and insomnia significantly worsened the affective, but not the sensory, pain component or pain intensity. The link between the affective (not the sensory) dimension of pain and both anxiety and insomnia, and the additional association found between anxiety and insomnia, completed an “affective pain–insomnia–anxiety” loop, while pain intensity did not uncover meaningful interactions. These results support our hypothesis, reinforcing the importance of considering pain as a multidimensional concept in the study of chronic pain, especially in FM research.

The affective component of pain is evaluated through items that give the emotional valence to pain but are at the same time evocative of anxiety patterns, even if they do not properly evaluate anxiety [45,53]. Thus, it is not surprising to find a correlation trend between the affective component of pain and anxiety, and most importantly, high affective pain scores in anxious FM patients. As for the link between insomnia and affective pain, one possible explanation could be the disruption in the emotional modulation of pain induced by insomnia [54], related to impaired activities in brain regions such as the amygdala, anterior cingulate cortex (ACC), insula, or frontal cortex [55], which are also implicated in affective pain processing [28]. Interestingly, these brain areas undergo function alterations in FM patients [56] and can be experimentally influenced by negative mood [57]. As brain pathways and structures regulating the affective dimension of pain are also implicated in anxiety [58] and sleep disorders [55], they probably constitute one important basis for the relationship between the affective pain dimension and both insomnia and anxiety.

Anxiety was associated with insomnia in FM patients, but insomniac patients tended to be more anxious than the anxious patients were insomniac, which suggests that insomnia more importantly influenced the appearance of anxiety than the reverse. Since both anxiety and insomnia similarly impacted affective pain, one plausible hypothesis about the “affective pain–insomnia–anxiety” loop could be that anxiety acts as a mediator of the link between insomnia and affective pain, as supported by our results. Indeed, anxiety has been shown to mediate the relationship between insomnia and pain incidence [59], while negative and positive affects mediate the relationship between sleep and pain interference [60].

Although the relationship between sleep disorders and chronic pain has often been reported as bidirectional, recent findings suggest the direction to be stronger from sleep disorders to chronic pain [11]. Thus, the loop would preferentially orient in the direction from insomnia to pain and to anxiety, and secondarily go from anxiety to pain (Figure 3). This statement has important clinical implications, because treatments aiming at reducing insomnia could at the same time reduce anxiety, and further decrease pain. Indeed, such reports already exist in the literature. For instance, a recent study found that a cognitive behavioral therapy (CBT) targeting insomnia not only improved sleep quality, but also showed a better long-term improvement in both sleep and pain than a CBT targeting pain, although the latter also induced some sleep amelioration [61]. Furthermore, a CBT targeting both pain and sleep dysfunction was more effective in reducing insomnia than a CBT targeting only pain [62].

Nevertheless, pain increases/induces anxiety, which in turn, worsens pain perception [19]. In addition, pain is known to negatively affect sleep, inducing insomnia among other sleep dysfunctions [63]. These observations maintain the bidirectional relationship, despite the preferential direction mentioned above (Figure 3). Overall, they illustrate how, in FM, restoring one component of the abovementioned pain–insomnia–anxiety loop could, through a sort of virtuous circle, allow improvement of the other components.

One mechanism underlying the pathological vicious circle between pain, insomnia, and anxiety could be the hyperexcitability of the central nervous system that exists in chronic pain syndromes and to which highly contributes central sensitization, one of the main FM features [64]. Indeed, insomnia, anxiety, and chronic pain are each associated with increased brain excitation [65,66], and could thus mutually reinforce each other this way. The implication of limbic structures (see above [55,58]) could explain the importance of the affective pain component in the loop. However, these highly speculative allegations need confirmation through dedicated investigations, in order to give more insights into this clinical loop disclosed as a potential key player in the pathology of FM.

Depression was highly prevalent among FM patients. However, apart from the well-known correlation with anxiety [67], depression was solely associated with sensory pain, pain interference, and insomnia, and this was true only in the presence of high ongoing pain intensity. Pain extent, measured with the WPI, was also correlated to ongoing pain. Interestingly, pain extent is associated with depression severity in FM patients [68]. Overall, these findings suggest that depression could be a direct consequence of suffering due to sensory painful feeling in FM patients when the daily level of pain (and thereof pain extension) was particularly high. Thus, depression does not appear to be part of the pain–insomnia–anxiety pathological loop. Nevertheless, one cannot completely rule out a possible participation of depression in FM pathology, since depression and FM share similar physiological mechanisms and genetic predisposition, especially in the case of major depressive disorder [33].

The sensory component of pain was related to the neuropathic feature and the severity of pain, which both correlated also with the general pain level. These results point out the contribution of the sensory dimension of pain in the disease burden, in addition to associated depression. However, the lack of correlation with descriptors of mood and sleep disorders suggests that it plays a minor role in the interplay between pain, sleep, and mood dysfunctions, or has only an indirect link through correlation with the affective pain component. The latter was strongly correlated with pain interference, which, as expected from the pathological loop hypothesized above, was also associated with insomnia and, to a lesser extent, to anxiety. Altogether, these observations further discriminate how the sensory and the affective pain dimensions, and associated symptoms and pain modalities, would, respectively, contribute to the high impact of FM on the patients’ quality of life [2,69].

High ongoing pain, by strengthening the association between anxiety and insomnia, and disclosing a link between sensory pain, insomnia, and depression, acted as an amplifier reinforcing the affective pain–insomnia–anxiety loop (Figure 3). In line with these findings, a recent study showed a prominent role of ongoing pain intensity in FM [48], with increased functional connectivity between the default-mode network and the insula—a key structure in pain regulation—a cluster noticed in the anterior insula highly implicated in the control of the affective dimension of pain [28]. Alteration in functional connectivity could be a marker of central sensitization [70], a prominent pathological feature in FM [71]. Thus, the deleterious role played by ongoing pain in FM appears to originate from functional impairment within the pain matrix.

Despite interesting findings, this study holds some limitations, the most important of them being the small sample. Additionally, correlation trends, instead of statistically significant correlations for some associations (despite shown BF), impose cautiousness in confirming the proposed pathological clinical model. The affective pain–insomnia–anxiety loop should not be considered as displaying a causal relationship between studied variables, which was beyond the scope of the cross-sectional study design.

## 5. Conclusions

This study, by pointing out the prominent role of the affective dimension of pain and its association with insomnia and anxiety in FM, proposes for the first time a clinical pathological model involving all three clinical indicators, with potential therapeutic implications. This model, while suggesting affective pain as a key pain dimension in FM, constitutes an additional indication that FM should be approached through a multidimensional scope when attempting to understand underlying pathological mechanisms or when seeking appropriate therapeutic strategies. The amplifying role of ongoing pain level by integrating external factors to the model or by strengthening associations between model components is in accordance with its recently shown impact on dysfunctional connectivity in pain-related pathways of FM patients. Despite the small sample size, the disclosed affective pain–insomnia–anxiety loop fits in a theoretical frame coherent with existing literature and deserves further confirmation through larger and more documented studies. Ultimately, this study points out the importance of focusing on meaningful clinical variables and outcomes of pain syndromes in order to better understand their pathological determinants and optimize their management.

## Figures and Tables

**Figure 1 jcm-11-03296-f001:**
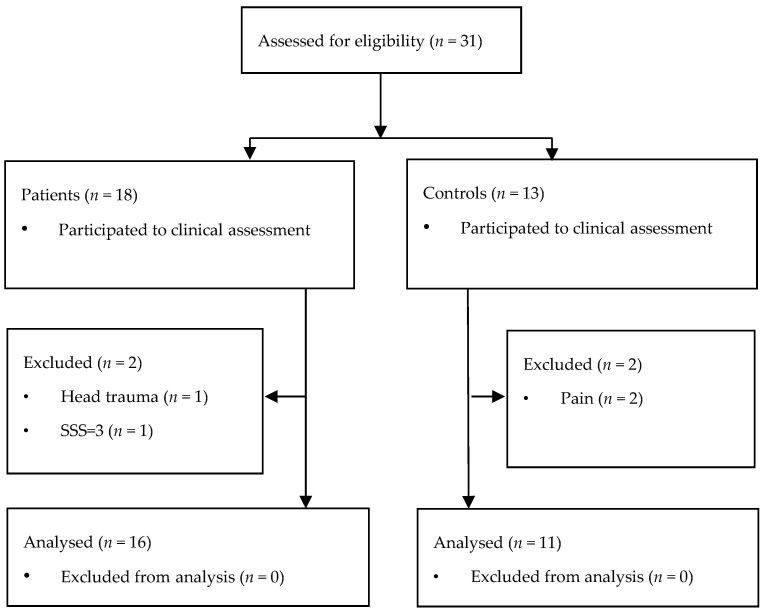
Overview of the selection procedure for participants in the study. In total, 31 participants were screened (18 patients and 13 controls). Two patients were secondarily excluded from the analysis because they finally presented one exclusion criterion each (head traumatism and one of the FM diagnosing score below defined limit). The two excluded controls complained of pain when they were interviewed.

**Figure 2 jcm-11-03296-f002:**
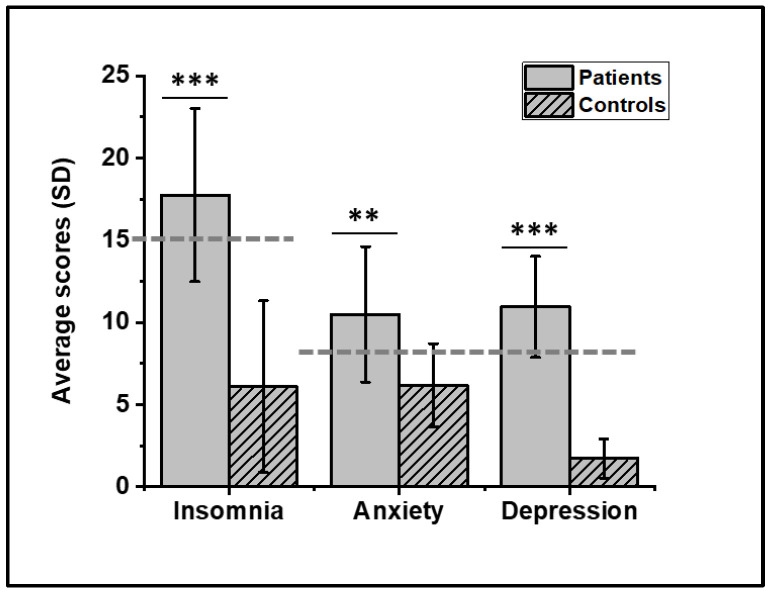
Sleep and mood scores in FM patients and controls. Dashed horizontal lines represent respective threshold scores for clinically significant insomnia (ISI ≥ 15), anxiety, and depression (HADanx and HADdep ≥ 8). Different levels of significance are represented, respectively, by ** (*p* < 0.01) and *** (*p* < 0.001). The ISI average score disclosed moderate insomnia in FM patients (17.75(5.27)) and was significantly higher than controls who had no insomnia (6.09(5.22), *p* < 0.001). Patients were at the limits of moderate anxiety (10.50(4.12)) and depression (10.94(3.07)), while controls had normal scores (respectively, 6.18(2.52), *p* = 0.005; 1.71(1.19), *p* < 0.001).

**Figure 3 jcm-11-03296-f003:**
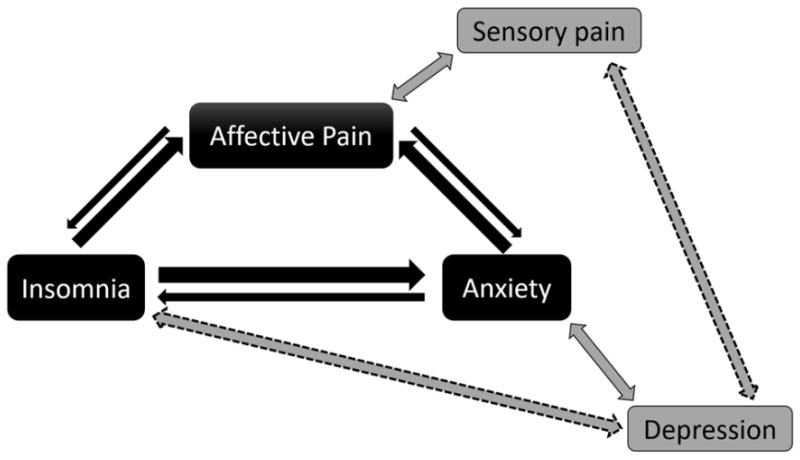
Affective pain–insomnia–anxiety loop and the role of ongoing pain intensity. The main constituents of the loop are labeled in black: affective pain, insomnia, and anxiety. Similarly, black arrows show their relationship, with the main direction being indicated by thicker arrows. Insomnia directly relates to affective pain but also through anxiety. Thus, all main arrows point to affective pain. The reverse relationship, although existing, is not favored by currently available data (narrower black arrows). Sensory pain is correlated to affective pain, and not to insomnia or anxiety, while depression is solely related to anxiety. These links seem to be out of the loop (grey boxes and arrows). In the presence of significant ongoing pain (VAS ≥ 3), not only does the correlation between insomnia and anxiety increase (not shown), but a new correlation appears on one hand between sensory pain and depression, and on the other hand between insomnia and depression (dashed grey arrows). Thus, high pain appears not only to reinforce the loop, but also to put into play other clinical factors that primarily appeared not to be involved in the loop (i.e., depression and sensory pain).

**Table 1 jcm-11-03296-t001:** Pain-associated neurological symptoms and treatment in FM patients.

**Neurological Symptoms**	**Type of Symptoms**	***n*/16**
	Negative symptoms *	7
	Positive symptoms **	4
	Normal	5
**Treatment**	**Type of Treatment**	***n*/16**
	NSAID	7
	Antidepressants	7
	Physical and alternative	5
	Antimigrainous	4
	Benzodiazepines	3
	Opiates	2
	Other drugs	5
	None	3

* Hypoesthesia (touch and pain); ** Paresthesia and hyperesthesia. NSAID: Nonsteroidal Anti-Inflammatory Drugs.

**Table 2 jcm-11-03296-t002:** Correlations between clinical scores in FM patients.

	1	2	3	4	5	6	7	8	9	10
1. ISI	-									
2. HADanx	**0.59 ^a^***	-								
	**(0.84 ^a^**)**									
3. HADdep	0.41	**0.56 ^a^***	-							
	**(0.67 ^b^*)**	**(0.68 ^b^*)**								
4. MPQaff	*0.48 ^b^*	*0.44 ^b^*	0.14	-						
	(0.45)	(0.36)	(0.49)							
5. MPQsen	0.33	0.16	0.01	**0.64 ^a^****	-					
	(0.29)	(0.39)	**(0.73 ^a^*)**	**(0.69 ^b^*)**						
6. VASd	−0.37	−0.29	−0.19	−0.11	0.24					
	(0.27)	(0.43)	(0.26)	*(0.61 ^b^)*	(0.53 ^b^)					
7. VASgen	−0.00	0.02	0.15	0.29	**0.62 ^a^***	0.39	-			
	(0.27)	(−0.07)	(0.41)	(0.47)	(0.55 ^b^)	(0.18)				
8. BPIsev	0.22	0.15	0.07	**0.51 ^b^***	**0.71 ^a^****	*0.44 ^b^*	**0.66 ^a^****	-		
	(0.40)	(0.33)	(0.50)	**(0.68 ^b^*)**	*(0.63 ^b^)*	**(0.77 ^a^*)**	**(0.66 ^b^*)**			
9. BPIint	**0.63 ^a^****	*0.47 ^b^*	0.39	**0.64 ^a^****	0.29	−0.41	−0.03	0.26	-	
	(0.55 ^b^)	(0.38)	**(0.69 ^b^*)**	**(0.72 ^a^*)**	(0.43)	(0.15)	(0.36)	(0.44)		
10. WPI	−0.22	−0.07	−0.36	−0.36	0.01	**0.57 ^a^***	0.22	0.18	−0.40	-
	(0.29)	(0.42)	(0.55)	(−0.46)	*(−0.58 ^b^)*	(0.02)	*(−0.66 ^b^)*	(−0.35)	(−0.38)	
11. DN4	−0.10	−0.02	0.12	**0.55 ^b^***	**0.71 ^a^****	0.34	**0.69 ^a^****	**0.63 ^a^****	0.20	−0.10
	(0.35)	(0.04)	*(0.65 ^b^)*	(0.47)	**(0.67 ^b^*)**	(0.11)	**(0.90 ^a^**)**	*(0.59 ^b^)*	(0.47)	*(−0.66 ^b^)*

Correlation coefficient (Pearson’s r) of the subgroup of FM patients with VAS ≥ 3 (*n* = 9) are written in brackets below the coefficient of the whole group (*n* = 16). Significant correlations are in bold (* *p* < 0.05, ** *p* < 0.01), while correlation trends (0.1 > *p* ≥ 0.05) are in italic. ^a^ BF ≥ 3. ^b^ 1 < BF < 3.

**Table 3 jcm-11-03296-t003:** Univariate comparisons of pain, sleep, and mood scores in FM patients between pathological and nonpathological groups.

		N	Mean ± SD	F	*p*-Value
Affective pain					
	No insomnia	5	3.00 ± 1.59	13.36	**0.003**
	Insomnia	11	7.18 ± 2.30		
	No anxiety	6	3.58 ± 2.31	9.71	**0.008**
	Anxiety	10	7.25 ± 2.26		
Sensory pain					
	No insomnia	5	4.11 ± 2.03	1.99	0.180
	Insomnia	11	5.83 ± 2.34		
	No anxiety	6	4.40 ± 2.11	1.45	0.248
	Anxiety	10	5.83 ± 2.39		
Current pain intensity					
	No anxiety	6	5.17 ± 2.18	0.20	0.665
	Anxiety	10	4.5 ± 3.26		
	No insomnia	5	5.90 ± 2.22	1.21	0.290
	Insomnia	11	4.23 ± 3.03		
General pain intensity					
	No anxiety	6	6.08 ± 2.51	0.17	0.683
	Anxiety	10	6.55 ± 1.94		
	No insomnia	5	6.00 ± 2.21	0.22	0.647
	Insomnia	11	6.54 ± 2.14		
Insomnia					
	No anxiety	6	14.83 ± 5.67	3.41	**0.086**
	Anxiety	10	19.50 ± 4.40		
Anxiety					
	No insomnia	5	6.60 ± 1.82	10.80	**0.005**
	Insomnia	11	12.27 ± 3.61		
Depression					
	No insomnia	5	9.20 ± 2.95	2.58	0.130
	Insomnia	11	11.73 ± 2.90		

*p* values corresponding to significant differences (*p* < 0.05) or significant trends (0.1 > *p* ≥ 0.05) between means are in bold.

## Data Availability

The data presented in this study are available on request from the corresponding author.

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
