# Peer review of "The Affective Dimension of Pain Appears to Be Determinant within a Pain–Insomnia–Anxiety Pathological Loop in Fibromyalgia: A Case-Control Study"

_jcm, 2022, doi:10.3390/jcm11123296_

Round 1
Reviewer 1 Report
This interesting manuscript describes a study which aimed to “investigate how pain (evaluated through its affective and sensory dimensions, as well as other indictors), mood and sleep disorders would interact in FM patients.” It was based upon the complex relationships among pain, sleep and mood disorders among persons with fibromyalgia. The authors note that sensory and affective pain stem from different neuroanatomical processes and are associated with different clinical outcomes. They state that better understanding these interrelationships “would not only give new insights in elucidating pathological processes engaged in FM but would also further enlighten underlying neurophysiological and structural mechanisms.” The study is part of a larger one that is looking at the involvement of brain GABAergic signaling in FM.
The major contributions of the paper are the findings related to sensory vs. affective pain interactions with sleep and mood disorder measures, and the finding that in the absence of current pain > 3 (visual analog scale, a commonly used clinical metric), some of the interactions with sleep and mood were suppressed. Figure 1 shows a model that could be used by clinicians to drive care delivery. The finding that higher current pain (VASd) seemed to amplify anxiety and sleep problems in FM points to the possibility that some in the FM sample were getting appropriate treatment or using self-management strategies that were effective in diminishing symptoms (e.g., low pain scores and less comorbidity).
The study was done under appropriate international ethical standards. The design is appropriate with appropriate inclusion/exclusion criteria for both the clinical and the control samples, commonly used measures, and appropriate statistical analysis. The figures and tables are appropriate and add to the write up. The authors' conclusions are stated appropriately for findings with a very small sample; the tentativeness of the findings might be further emphasized by some sort of caveat in the first Discussion paragraph. The authors do a good job explaining the potential neurophysiologic ramifications of their findings.
The manuscript is easy to follow but there are several grammatical errors and awkward statements throughout.
References the authors might want to consider using in their discussion are those examining treatment for insomnia vs. pain in FM such as the following
McCrae CS, Williams J, Roditi D, Anderson R, Mundt JM, Miller MB, Curtis AF, Waxenberg LB, Staud R, Berry RB, Robinson ME. Cognitive behavioral treatments for insomnia and pain in adults with comorbid chronic insomnia and fibromyalgia: clinical outcomes from the SPIN randomized controlled trial. Sleep. 2019 Mar 1;42(3):zsy234. doi: 10.1093/sleep/zsy234.
Prados G, Miró E, Martínez MP, Sánchez AI, Lami MJ, Cáliz R. Combined cognitive-behavioral therapy for fibromyalgia: Effects on polysomnographic parameters and perceived sleep quality. Int J Clin Health Psychol. 2020 Sep-Dec;20(3):232-242. doi: 10.1016/j.ijchp.2020.04.002.
Reviewer 2 Report
This is a very interesting and well written study, albeit the small sample of patients-controls. It is overall well conducted and well explained, allowing replications, and its findings are quite interesting in light of recent pain research. A couple of minor points to be addressed follow.
- Unfortunately, some references, especially in the introduction, are a bit outdated. I would suggest, for example, to substitute ref. 22 with Salaffi, Fausto, Piercarlo Sarzi-Puttini, and Fabiola Atzeni. "How to measure chronic pain: new concepts." Best Practice & Research Clinical Rheumatology 29.1 (2015): 164-186.
- I think that the explanation of why patients and controls were chose as right-handed should be also put in the "participant" section, where the EEG study is cited (ref. 32) without explanation (which however can be found further on in the text).
